# Stimuli-Responsive Polymeric Nanomaterials for the Delivery of Immunotherapy Moieties: Antigens, Adjuvants and Agonists

**DOI:** 10.3390/ijms222212510

**Published:** 2021-11-19

**Authors:** Raveena Nagareddy, Reju George Thomas, Yong Yeon Jeong

**Affiliations:** 1Department of Biomedical Sciences, Chonnam National University Hwasun Hospital, Hwasun 58128, Korea; rraveena1996@gmail.com; 2Department of Radiology, Chonnam National University Hwasun Hospital, Hwasun 58128, Korea; regeth@gmail.com

**Keywords:** stimuli, polymer nanoparticle, tumour microenvironment (TME), cancer immunotherapy

## Abstract

Immunotherapy has been investigated for decades, and it has provided promising results in preclinical studies. The most important issue that hinders researchers from advancing to clinical studies is the delivery system for immunotherapy agents, such as antigens, adjuvants and agonists, and the activation of these agents at the tumour site. Polymers are among the most versatile materials for a variety of treatments and diagnostics, and some polymers are reactive to either endogenous or exogenous stimuli. Utilizing this advantage, researchers have been developing novel and effective polymeric nanomaterials that can deliver immunotherapeutic moieties. In this review, we summarized recent works on stimuli-responsive polymeric nanomaterials that deliver antigens, adjuvants and agonists to tumours for immunotherapy purposes.

## 1. Introduction

Immunotherapy is a hopeful method for the treatment of various diseases, including autoimmune diseases, cancers and some infections. Cancer immunotherapy is an emerging field of medicine where the patient’s own immune cells are used to treat the tumour [1,2,3,4,5,6]. Cancer immunotherapy treats cancer by inducing a strong antitumour immune response and plays important roles in controlling metastasis and preventing recurrence, hence set forth as a major advantage over long-established cancer treatments [7]. Stimulating immunity against cancer includes several key aspects, including the release of antigens from TME and its uptake by antigen-presenting cells (APCs), presentation of tumour antigens by APCs, priming and activation of T cells by activated APCs, migration and infiltration of effector T cells back into the tumour, and the recognition and killing of tumour cells by effector T cells [8,9]. To achieve these objectives, scientists have been exploring immunotherapy by targeting all types of immune cells individually, such as dendritic cells, macrophages, T cells, natural killer cells and myeloid-derived suppressor cells.

Nanotechnology plays an important role in drug delivery because nanomaterials can be modulated based on the intended purpose because of its high surface to volume ratio and wide range of purposes based on the material and size it is made of [10,11]. The main advantage comes because of its high therapeutic index over traditional medicine delivery and adjustable pharmacokinetics. Polymeric nanomaterials play an important role in drug delivery due to their biocompatibility and biodegradability [12]. Some polymers even have the ability to activate immunity [13]. In contrast, immunomodulatory drugs administered systemically have failed due to severe toxicity (because the body’s overall immunity is affected). Formulations of the same drugs in nanoparticle form have shown greatly increased localization in target tissues or within immune cells, thereby increasing their potency and enhancing their safety [14]. The polymeric nanoparticles that are commonly used in cancer immunotherapy are poly(g-glutamic acid) (PGA), poly(ethylene glycol) (PEG), poly(D,L-lactic-co-glycolic acid) (PLGA), poly(D,L-lactide-co-glycolide) (PLG), chitosan and polyethyleneimine (PEI) nanoparticles [15]. As previously mentioned, polymeric nanoparticles are widely used to deliver immunostimulatory agents because they exhibit excellent biocompatibility, biodegradability, high loading capacities for immune-related components, chemical stability and water solubility. The added advantage of polymers is their responsiveness to certain internal and external stimuli. Internal stimuli include pH (Potential of hydrogen), ATP(Adenosine triphosphate), H_2_O_2_ (Hydrogen peroxide), enzymes, redox potential and hypoxia, and external stimuli include magnetic fields, temperature (i.e., thermal), ultrasound, light (e.g., laser) and electronic fields [16]. Stimulation could occur in the TME (Tumour microenvironment) or inside cancer cells.

The major challenge faced in the immunotherapy field is to induce a specific immune response [17] by triggering naive T cells directly or by activating APCs in order to subsequently present antigens to CD8+ and CD4+ T cells [18]. To optimize immunotherapy, the system will need an antigen, an adjuvant and optionally an inhibitor or agonist [19].

Stimuli-responsive polymers are designed specifically to release drugs, antigens, adjuvants or agonists in a particular area, where the pathological profiles are different from the normal profile of a tissue [20]. Similarly, nanoparticles designed to release drugs due to exogenous stimuli, such as light, acoustics, temperature and magnetic or electric fields, appear to have more control over drug release. In this review, we focus on introducing stimuli-responsive polymeric nanomaterials as carriers for the effective delivery of cancer antigens, adjuvants and agonists for cancer immunotherapy.

## 2. Stimuli-Responsive Polymeric Nanomaterials

### 2.1. Endogenous Stimuli

A tumour is generated when several mutational changes occur in cells that escape encounters by the body’s immune system and cell signalling pathways. These mutational changes cause gene amplification that make the cells produce mutated proteins such as receptor tyrosine kinases (RTKs: e.g., EGFR), Serine/threonine kinases (e.g., Akt), lipid kinases (e.g., PI3Ks), etc., [21]. All these changes occur at the cellular level, which is consistent over the tumour growth. For example, the RTKs activate the PI3K which will in turn activate the Akt pathway and draw more glucose molecules inside the cancer cell [22]. The end result of most mutations will be either high uptake of glucose or an increase in pH, glutathione, reactive oxygen species (ROS), certain enzymes and hypoxia, which are all related factors (Figure 1).

As the tumour grows, in order to oblige the new mass of cells, new irregular neoangiogenesis occurs, which reduces the oxygen content in the tumour environment [23]. Thus, cancer cells undergo anaerobic glycolysis, accumulating lactic acid and resulting an acidic environment [24]. The bonds in some polymers can be degraded at acidic pH due to protonation and deprotonation of ionizable groups [25]. Lang et al. developed pH-sensitive carboxy-dimethylmaleic amide-linked methoxy poly (ethylene glycol)-*b*-[poly (diisopropylamino) ethyl methacrylate] nanoparticles with free amine side chains that break around pH 6.5 for the delivery of ovalbumin to enhance antitumour immunotherapy [26]. Polymeric materials that are responsive to various endogenous stimuli are listed in Table 1.

GSH (Glutathione) has an exceptional influence on many cell functions, including cell cycle regulation, gene expression, immune responses, protein function and cell death activation [38]. The concentration of GSH in the cellular cytoplasm is twice or thrice higher in magnitude than that in the extracellular matrix [39]. The GSH level in tumour tissues is much higher than that in normal tissues. Disulphide bonds reduce to thiol groups in a reducing environment such as TME with high GSH. Shen et al. developed a mPEG-PPLG-based nanoparticle that is GSH-responsive for the release of DOX(Doxorubicin) and SO_2_ (Sulfur dioxide) as prodrugs [40].

ROS refer to a group of oxidative molecules, such as H_2_O_2_, OH- and superoxide ions, produced in a cell [41]. When produced in excess, ROS are linked to many metabolic reactions that are responsible for protein synthesis, proliferation, tumorigenesis, etc. [42]. Generally, tumour tissue presents extremely high levels of intracellular oxidative stress compared with healthy tissue that indirectly increases H_2_O_2_ levels and thus ROS [39]. The differences in cancer cells that act as endogenous stimuli are summarized in Figure 1. The basic metabolism is disrupted and uncontrolled, thus leading to unbalanced production of proteins and enzymes. Shim et al., synthesized poly(amino thioketal), an ROS sensitive polymer that cleaves the thioketals for the gene delivery [43].

Enzymes are proteins that have defined functions. If the transcription and translation of a cell are faulty, the balance of enzymes is lost, which leads to the high concentration of a particular enzyme [44]. Certain enzymes will show high levels in certain types of cancers, and a nanocarrier can be generated so that it is cleavable by that enzyme. Guan et al., constructed sulfato-b-cyclodextrin nanoparticles and the anticancer choline-modified prodrug chlorambucil, and the nanoparticles are cleaved by the enzyme butyrylcholinesterase, which releases the anticancer drug chlorambucil [36].

Some regions of solid tumours grow uncontrollably in a hypoxic state because of irregular angiogenesis and high cellular proliferation rates [45]. Cancer cells gravitate towards oxygen-rich locations, which leads to metastasis, and these regions show high drug resistance since basic metabolism is dysregulated [45]. The acceptance of electrons by polymers under hypoxic conditions can alter the particle size and hydrophilicity of certain polymers, such as nitrobenzoyl alcohols, nitroimidazoles and azo linkers [37,46]. For example, Son et al., developed carboxymethyl dextran–black hole quencher 3 nanocarrier (has azo bonds) for the release of DOX. The azo bonds reduce under hypoxic conditions for the release of DOX [47].

### 2.2. Exogenous Stimuli

Various kinds of polymeric materials are capable of responding to various external stimuli, such as light, temperature, ultrasound and magnetic fields (Table 2). Remote activation helps achieve the timely triggering of multiple nanocarriers to release drugs. Figure 2 shows nanoparticles loaded with immunomodulatory agents or drugs taken up by tumours and released in the TME with the help of exogenous stimuli.

Irradiation is one of the most widely used stimuli because of its biocompatibility, ease of handling and cost effectiveness. The irradiation uses ranges from visible to ultraviolet (UV) to near infrared (NIR) [57]. UV light can cause phase transitions in polymers with special structures, such as O-nitrobenzyl, pyrene, azobenzene and spiropyran [48,49,50,51]. Visible light (Vis) with a longer wavelength is more permeable in tissues, is widely used in photothermal therapy (PTT) and photodynamic therapy (PDT) and is safer for clinical applications. NIR has better penetrability than Vis and UV, thus leading to increased therapeutic efficacy [58]. Son et al., synthesized hyperbranched polyglycerol micelles using spiropyrans; the spiropyrans will isomerize to merocyanine upon UV exposure and release hydrophobic drugs[59].

Thermosensitive polymers can act as heat-sensitive materials for controlled drug release, and their solubility can change at transition temperatures [60]. Transition temperatures are the lower critical solution temperatures (LCST) and upper critical solution temperatures (UCST) of a solution at which transition occurs. Thermosensitive nanoparticles are formulated using heat-responsive polymers and drug conjugates, and they release drugs at the point of critical temperature [60,61]. For example, PNIPAm an amphiphilic polymer has an LCST at 32 °C; when the LCST is passed, the material hides its hydrophilic sites and exposes the hydrophobic areas, and the process is reversible. Chung et al., made polymeric micelles with pNIPAm and poly(butylmethacrylate) for the delivery of drug adriamycin(DOX) [62].

Ultrasound (US) is a stimulus that can penetrate deeper into the body and activate nanovehicles without damaging normal healthy tissues [63]. It can also trigger the release of drugs through mechanical and thermal stimuli generated by the phenomenon of cavitation and acoustic radiation force [64]. Cao et al., developed two nanomaterials one made of lipids and the other with PLGA for the delivery of perfluoropentane and DOX using low intensity focused ultrasound. The two nanomaterials required two different parameters, Lipid nanodroplets were releasing drugs at 3 W/3 min and PLGA nanodroplets were releasing at 8 W/3 min (at which phase transition occurred) and the survival rate of the animals was better when the number of pulses was high rather than a single pulse [65].

Magnetically responsive materials loaded inside polymers respond to alternating magnetic fields or static magnetic fields that generate heat and mechanical stress to release the loaded moieties (drugs) [66]. These nanoparticles can also be pulled along a magnetic field to the tumour site. Most magnetic materials also act as good contrast agents for MRI [67]. Cortes et al. synthesized nanogels with OEGMA and MAA loaded with superparamagnetic iron oxide, thus creating a gel that was responsive to magnetic, pH and thermal stimuli [55]. The electric field is another emerging method of triggering drug release [68], although limitations in applying electric fields in vivo have limited the use of this method.

## 3. Stimuli-Responsive Polymers for Antigens, Adjuvants and Agonists

### 3.1. Antigens

Antigens are molecular structures composed of proteins that can initiate a cascade of immune reactions in the system [69]. The antigens in the system will be recognized by the wandering immune cells such as macrophages and dendritic cells and uptaken via endocytosis. The uptaken antigens by the immune cells, say DCs will lyse the antigen to present the epitope to other immune cells such as T cells. The maturation and presentation of the epitopes to immune cells happens in the lymph nodes. Most of the antigens used in cancer research are model proteins such as CPG (an oligonucleotide with 5’-C-phosphate-G-3’) and ovalbumin (a protein) that can stimulate a wide range of cells and cause a wide range of reactions such as induction of TNF-α, IL-6, IL-10, IL-12, IL-27.

#### 3.1.1. Endogenous Stimuli

Endogenous stimuli for the delivery of antigens include pH, redox reactions or hypoxic conditions (Table 3). pH-Sensitive polymers are dissociated by shifting the carrier charge, using acid liable linkages or using pH-responsive cross linkers [70]. Knight et al. developed the pH-responsive polymeric nanoparticles propylacrylic acid (PAA), butyl methacrylate (BMA) and dimethylaminoethyl methacrylate (DMAEMA) for the delivery of the ovalbumin antigen and CpG (TLR 9) adjuvant through intranasal injection to activate lung and resident T cells as a virus treatment. The antigen was held in particle due to pyridyl disulphide group present in the pyridyl disulphide ethyl methacrylate (PDSMA) and the cationic charge of DMAEMA holds the anionic nucleic acid adjuvant (CpG). After endocytosis, the PAA, BMA and DMAEMA copolymer destabilizes due to the pH of endosome releasing the antigens in the cytosol and the immunized mice showed a higher percentage of IFNγ and TNFα. [71]. Miyazaki et al. developed pH-responsive CD44-targeting HA nanoparticles for the delivery of a model antigenic protein (ovalbumin) to DCs. The nanoparticles reach the endosomes of the cells and are cleaved under the acidic pH (4.5) of the endlysosomes, thus releasing the model antigenic protein (Figure 3). Antigens are cleaved by acidic pH, and epitopes are presented to T cell receptors via MHC class I molecules, thus allowing cytotoxic T cells to recognize cancer cells [72]. Similarly, Eiji et al. developed pH-responsive 2-carboxycyclohexane-1-carboxylated dextran (CHex-Dex)-based nanoparticles for endosome cleavage and cytosolic delivery of antigenic proteins (OVA). They compared CHex-Dex with MGlu-Dex and proved CHex-Dex is a promising pH-responsive polymeric nanomaterial due to its improved hydrophobicity (enhanced cellular association) with both a cytoplasmic antigen delivery function and strong DC activation properties for the induction of cellular immunity [73].

pH-Sensitive polymeric micelles also deliver antigens to DCs that induce high CTL responses. Yang et al. developed chitosan-based micelles with ovalbumin as an antigen that targets lymph nodes [74]. Yoshizaki et al. developed 3-methyl-glutarylated hyperbranched poly(glycidol) (MGlu-HPG) with the antigen CPG [75]. Zhou et al. investigated pH-responsive nanovaccines to deliver the model antigen ovalbumin, and the vaccine activates the STING pathway of DCs using 5,6-dimethylxanthenone-4-acetic acid and includes poly (ethylene glycol)-block-poly [2-(diisopropanol amino) ethyl methacrylate] as a backbone [76]. Okubo et al. produced pH-sensitive chondroitin sulphate (CS)-derived liposomes for the delivery of the antigen ovalbumin to the endosomes of DCs [77].

Similarly, the polymer HPAA-F7 can be used as a vaccine carrier to induce potent antitumour cellular immunity. A fluorinated HPAA (HPAA-F7) was prepared using redox-responsive fluorinated hyperbranched polyamidoamine (HPAA) with heptafluorobutyric anhydride [79]. Selective delivery of oncolytic viruses to cancer cells was achieved by Deng et al. They developed a redox-sensitive HA-based nanohydrogel (cleavage of disulphide bonds) for the delivery of oncolytic viruses that replicate and specifically destroy cancer cells [82]. Hydrogels are materials that are capable of holding and releasing drugs for a longer time, and they can maintain different shapes.

#### 3.1.2. Exogenous Stimuli

Based on the penetration of light through skin, NIR penetrates the deepest and can be easily externally activated. Zhang et al. produced an NIR-sensitive nanoparticle grafted with pheophorbide A (a hydrophobic photosensitizer) and polyethyleneimine (PEI) for the delivery of ovalbumin to DCs to enhance antigen specificCD8+ T cell response. Pheophorbide A coupled with PEI through carbodiimide coupling and antigen interacted with polymer electrostatically. The in vitro experiments showed that photosensitizer when irradiated at 670 nm (NIR) caused photochemical internalization mediated endosomal delivery of antigens to DCs [83]. Another photosensitizer, Ce6(negatively charged)-doped mesoporous silica nanoparticle with a hypoxia-sensitive azo linker and PEG that can hold a glycol chitosan CpG complex was developed, and upon reaching the hypoxic tumour regions, the two nitrogen bonds present in the azo linker break and release the model antigen CpG [81] (Table 4).

### 3.2. Adjuvants

Adjuvants are drugs or chemicals used to enhance the immune reaction of vaccines and antigens by generating cascading immune reactions that cause polarization or antibody release to the tumour sites [84]. As explained in the previous section, antigens are capable of causing an immune reaction, whereas as adjuvant can attenuate the same immune reaction to a higher extent. For example, Alum (TLR 7 adjuvant), used as a vaccine adjuvant for more than a century, induces a type 2 inflammatory response by increasing the MHC II levels and IL-4 [85]. Researches have shown that alum enhances the effects of CD4 and CD8 T cell priming and also enhances the antibody reactions [86]. After the activation of T cells, T cell death occurs because of the induction of activation induced cell death within few days, and some of the surviving T cells become memory T cells (long- life) that are capable of tackling antigen within hours. Other adjuvants such as CpG and some plasmids also enhance immune reactions via TLRs.

#### 3.2.1. Endogenous Stimuli

Polymers can be used to deliver adjuvants (Table 5). Kawai et al. reported the use of pH-responsive lipid nanoparticles encapsulating plasmid DNA, and Simon et al. developed amphiphilic block copolymers (*N*-[2-[(tetrahydro-2H-pyran-2-yl) oxy] ethyl]–acrylamide (HEAmTHP)) for delivering amphotericin B (a TLR 2 and TLR 4 activator). The former used CpG-free mcs (noncoding plasmid DNA) as an adjuvant that activates cytoplasmic DNA sensing pathways, such as STING/TBK1/IRF3 leading to increase in IFN- β and MHC molecules (I and II (3.1- and 1.3-fold, respectively), and costimulatory molecules (CD80, CD86 (2.1- and 1.7-fold, respectively)) in RAW 264.7 cells proving the adjuvant property. When they combined their treatment in vivo with anti-PD-1 therapy and nanoparticles, and they achieved lower tumour volumes relative to treatments that applied nanoparticles and anti-PD-L1 antibodies separately [87,88].

Liqin et al. produced ROS-sensitive polymers to encapsulate Ce6 and DOX and assessed the results of their nanoparticles in combination with anti-PDL1 therapy. ROS-sensitive nanoparticles in combination with anti-PDL1 antibodies not only increased the survival rate but also inhibited secondary tumours at distant regions [89]. Weijing et al. also investigated the delivery of HPHH (photosensitizer) and DOX by polymers that are responsive to glutathione. The polymersomes released DOX and HPHH due to reduction of disulphide bonds. DOX inhibits topoisomerase II (essential for DNA replication) and HPHH increases ROS upon and cause cell death along with release of TAA. The polymersomes itself acted as adjuvants and activated DC’s (up to 17.6%) because of the presence of tertiary and primary amines (Figure 4) [31]. Irradiation with lasers caused photosensitizers to generate ROS and initiate ICD (release antigens), and the antigens released from the tumour activate APCs to enhance the immunotherapeutic effect along with chemotherapy using DOX.

#### 3.2.2. Exogenous Stimuli

As an alternative to endogenous stimuli, exogenous stimuli are designed to release adjuvants and kill cancer by promoting antigen release for the immune reaction. Ultrasound is a harmless energy stimulus that can reach deeper tissues. Meng et al. reported ultrasound-responsive hydrogels made of oligo (ethylene glycol) methacrylate (OEGMA) and Laponite clay loaded with nanovaccines composed of PLGA nanoparticles with imiquimod as an adjuvant [53]. Li et al. developed polymer hybrid micelles for delivering CpG oligodeoxynucleotide adjuvants for cancer immunotherapy [90]. Meng et al. compared nanovaccine and nanovaccine-loaded hydrogels and showed that the nanogels led to stronger immune responses than three individual doses of the nanovaccine alone. Along with that, they studied the immune memory effect by Effector memory T cells percentage in the mice treated with Nanogels α-PD-1 combination proving strong antitumour responses [53].

### 3.3. Agonists

Agonists can be either a chemical or a biological material, such as a hormone or drug, and it binds to a receptor to attenuate the downstream effector mechanisms that produce a response [91]. Agonists such as poly I:C, dsRNA (TLR3 agonist) made synthetically or originated from virus, respectively, activates DC via TIR domain containing adapter inducing interferon-β (TRIF). TRIF can also result in the activation of NF-kB, ERK, JNK, etc. Similarly, CpG activates via the MyD88 adapter molecule. Retinoic acid-inducible gene I (RIG-I) [92]. Recently, agonists have been used as a main component in certain treatments.

#### 3.3.1. Endogenous Stimuli

Agonists are very similar to adjuvants and can attenuate the immune reaction. Kim et al. investigated pH-responsive PLGA nanoparticles for endolysosome-specific release of agonist 522 (TLR7/8 agonist). CO_2_ gas generated at acidic pH due to the sodium bicarbonate that was coloaded for better release and encapsulation of the TLR7/8 agonist 522, and the nanoparticles showed great tumour reduction [27]. These researchers investigated the same nanoparticles for the activation of NK cells via APCs and showed stronger in vivo cytotoxicity and prolonged NK cell activation compared to that of soluble agonists. Additionally, the treatment significantly enhanced the antitumour efficacy of cetuximab (EGFR inhibitor) in mouse tumour models [93].

Delivery of TLR7/8 moieties by Smith et al. in PEG−PLA nanoparticles led to slower tumour growth, extended survival and decreased systemic toxicity in comparison to free TLR7/8a when used along with anti-PD-L1 checkpoint blockade [28]. Other researchers developed pH-sensitive PEG histamine-modified alanine to deliver let-7b and nucleic acids that act similarly to TLR 7 agonists to repolarize macrophages and activate DCs along with a cationic *Bletilla striata* polysaccharide that acts both as a targeting moiety and vector for transfection [29] and intracellular enzyme-sensitive beta glucuronidase tagged with PEGylated imidazoquinoline using ester bonds. The particles activate immune reactions against tumours by attracting more DCs to the lymph nodes [94].

Screening of 30 polymer dialects of mPEGblock-[DMAEMA-co-AnMA] by varying the number of carbons in the monomer methacrylate and identified the four best polymer formulations, for the delivery of 3pRNA for the activation of RIG-1. The carriers were designed specifically to enhance the delivery and activity of 3pRNA and showed the potential to advance the clinical development of RIG-I agonists and enhance the immunostimulatory activity of 3pRNA in multiple cell types. Novel polymeric carriers have been designed and optimized specifically to enhance the delivery and activity of 3pRNA [95]. This group also worked with the same agonist on another pH-responsive polymer, p(DMAEMA)-b-(DMAEMA-co-BMA-co-PAA), and assessed its immunotherapeutic effects in a colon cancer model [96]. In contrast, Saborni et al. investigated the induction of synthetic immunogenic cell death using cGAMP (STING agonist) to activate the STING pathway in cancer cells using a pH-responsive PLGA nanocarrier. They also combined the concept with the chemo drug DOX and anti-CTLA-4 therapy and claimed to induce highly immunogenic cancer debris that has the ability to increase anticancer adaptive immunity [97] (Table 6).

#### 3.3.2. Exogeneous Stimuli

The sensitivity of material to radiation varies. Gao et al. made polymeric nanoparticles with PEG, RGD peptide, doxorubicin and radiation-activable selenide for NK cell-based immunotherapy. The conversion of selenide to selenic acid under radiation also alters GSH levels, and RGD is an immunogenic peptide used to treat breast cancer. These authors found that the tumour inhibition rate was highest when radiation, nanoparticles and RGD peptide were used together [98].

A NIR II activatable semiconducting material based on a polymeric nanoagonist conjugated with Resiquimod induces immunogenic cell death. With the help of agonists, DCs are activated, and they then activate T cells in a cascade reaction, as shown in Figure 5. Cytotoxic T cell responses were doubled in a treatment that applied a laser in comparison to that without a laser [99] (Table 7).

### 3.4. Codelivery of Antigens and Adjuvants

Antigens and adjuvants being presented simultaneously to the same antigen-presenting cell will be a key for triggering sufficient immunological responses for cancer immunotherapy.

Therefore, codelivery of adjuvants and antigens provides better curative outcomes. However, most of the adjuvants are nucleic acids, peptides and nucleic proteins that are known for their instability in vivo, leading to the degradation of the adjuvant before reaching the target cells, thus impeding the efficacy of current cancer immunotherapy. To improve the efficacy, many drug delivery strategies that use polymeric micelles as a means to co-deliver antigens and adjuvants are under investigation (Table 8).

For the codelivery of antigens and adjuvants, Zhouqi et al. developed ultrasound-responsive hydrogels loaded with nanovaccines made of PLGA nanoparticles with ovalbumin as a model antigen and imiquimod as an adjuvant. Hydrogels made of oligo(ethylene glycol) methacrylate and laponite present the burst release of nanovaccines in the presence of ultrasound and show cytotoxic T cell responses [53]. Li et al. developed polymer hybrid micelles for delivering tyrosinase-related protein 2 peptide antigens and CpG oligodeoxynucleotide adjuvants for melanoma immunotherapy and observed tumour reduction in mouse models [90]. Amphiphilic diblock copolymer poly(2-ethyl-2-oxazoline)-poly(d,l-lactide) (PEOz-PLA) combined with carboxyl terminated-pluronic F127 forms mixed micelles for the codelivery of the model antigen ovalbumin and TLR 7 agonist CL264, which targets draining lymph nodes. These materials are specifically taken up by DCs, and the micellar structure cleaves inside the endolysosome due to low pH. The codelivery of this combination in E.G7-OVA tumour-bearing mice significantly inhibited tumour growth and markedly improved the survival of tumour-bearing mice [100].

## 4. Conclusions

Stimuli-responsive polymers have a good prospect in the field of immunotherapy for drug delivery and can be improved by optimizing the material and dosage. Several factors are considered for the development of immunotherapeutic stimuli-responsive nanoparticles. The first factor is the biocompatibility of the material, which is crucial for FDA approval and subsequent clinical application. The second factor is the ease of synthesis, which is important for scaling up the nanomaterial. Usually, nanoparticle synthesis consists of complicated reactions that are often not reproducible. Therefore, reproducibility and scaling up are critical factors for the successful rollout of a nanoparticle-based drug. Another factor is animal model selection for the immunotherapeutic action of nanoparticles.

Despite all the technological advancements made, immunotherapy is currently in its infancy. Standard treatment methods used in chemotherapy with proven results might not be the case with immunotherapy as patient-to-patient the result may vary. As in the case of glioblastoma multiforme, the treatment protocol established currently is a combination of temozolomide and radiation therapy. Even though the survival rate for glioblastoma is very low, chemotherapy provides a guaranteed effect provided patient MGMT methylation status is favourable. Now, an immunotherapy alternate for glioblastoma is under trial despite the low immunogenicity of glioblastoma. The result of this trial heavily depends on individual patient tumour microenvironment and health of the immune system. Therefore, the development of a stimuli-responsive system for immunotherapy should take into consideration this challenge.

In the future, research to maximize the advantages of stimuli-responsive NPs is required so that these materials can be used in cancer immunotherapy in clinical settings

## Figures and Tables

**Figure 1 ijms-22-12510-f001:**
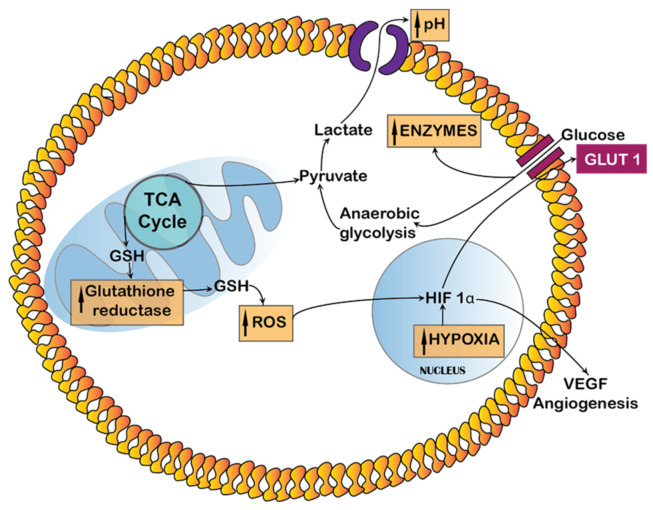
Endogenous stimuli responses that affect different regions inside cancer cells.

**Figure 2 ijms-22-12510-f002:**
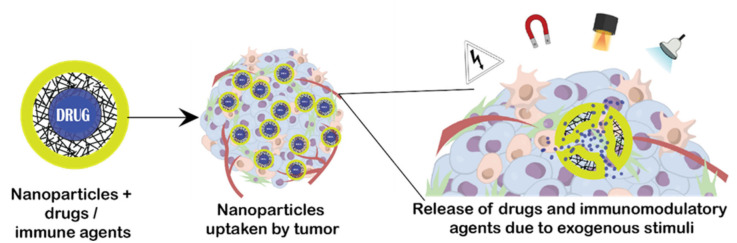
Different exogenous stimuli and their effects to release drugs or immunomodulatory agents in the TME.

**Figure 3 ijms-22-12510-f003:**
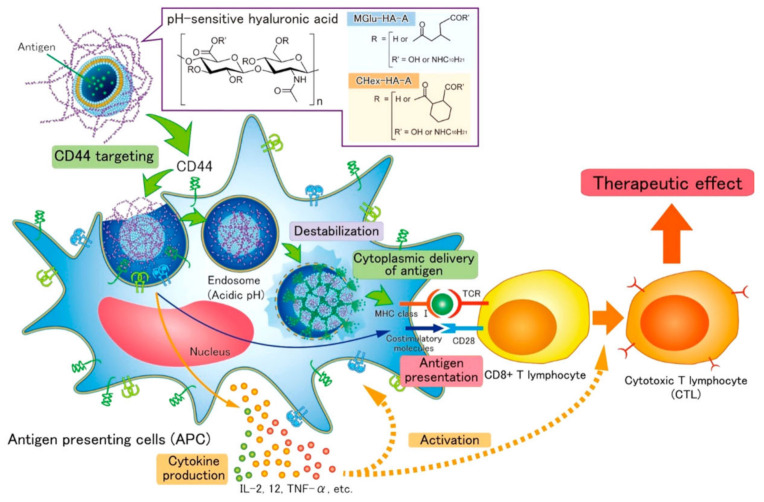
Schematic showing the delivery of antigen-OVA using liposomes with pH-sensitive CD44-specific delivery of hyaluronic acid derivatives [72]. Copyright ACS Biomater. Sci. Eng.

**Figure 4 ijms-22-12510-f004:**
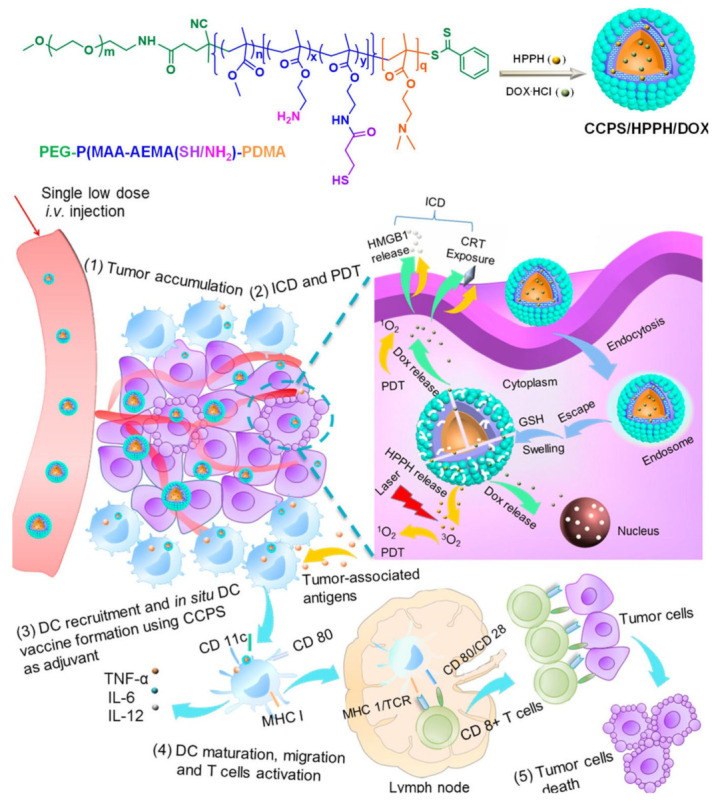
Delivery of adjuvants. Schematic illustration of a DC vaccine made of chimeric cross-linked polymersomes as an adjuvant and tumour-associated antigens induced by PDT and ICD for MC38 colorectal cancer immunotherapy. [31] Copyright ACS Nano.

**Figure 5 ijms-22-12510-f005:**
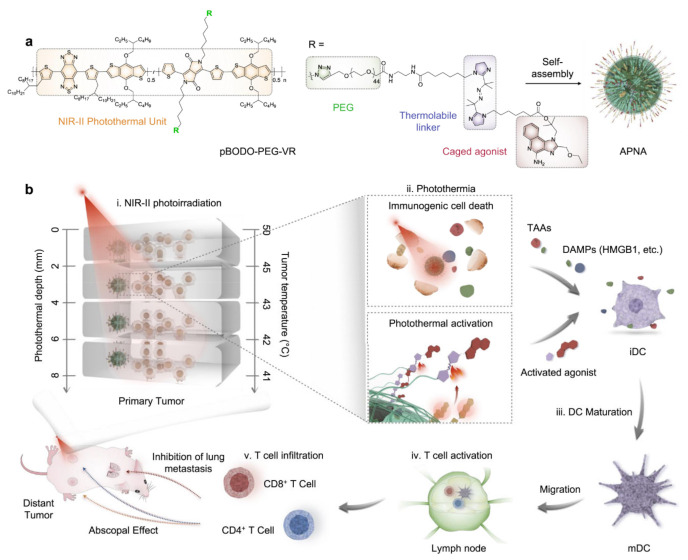
Delivery of agonists. (**a**) Schematic representation of APNA. (**b**) Mechanism of the antitumour immune response of the polymer APNA via NIR-II photothermal immunotherapy, in which tumour-associated antigens and damage-associated molecular patterns are released, activating DCs. [99] Copyright Nature Communications.

**Table 1 ijms-22-12510-t001:** Endogenous stimuli responsive polymeric nanoparticles.

Stimuli	Nanomaterials	References
pH	PAA, PMAA, PEI, PNIPAAM, PAM, PLGA, PEG histamine modified Alanine, PLA	[27,28,29]
GSH	Polymers with disulphide linkage PEG-P (MMA-co-AEMA (SH/NH2)-PDMA	[30,31]
ROS	Poly (propylene sulphide), poly(thioketal), phenyl boronic acid, poly- L-(methionine), poly- L-(proline)	[32,33,34,35]
Enzymes	Sulfato-b-cyclodextrin	[36]
Hypoxia	Nitrobenzoyl alcohols, Nitroimidazoles, Azo linkers	[37]

**Table 2 ijms-22-12510-t002:** Exogenous stimuli responsive polymeric nanoparticles.

Stimuli	Nanomaterials	References
Light	O-nitro benzyl, pyrene, spiropyran, and azobenzene	[48,49,50,51]
Thermo	HPMA, PNIPAAM, PiPOx, Modified Poly acrylamides	[52]
Ultrasound	PLGA, tetrahydropyranyl groups	[53,54]
Magnetic	OEGMA and MAA loaded with superparamagnetic iron oxide	[55]
Electric	PVA, poly (acrylic acid-co-2-acrylsmido-2-methyl propyl sulfonic acid)	[56]

**Table 3 ijms-22-12510-t003:** Endogenous stimuli-based polymer nanoparticles for delivery of antigens.

Stimuli	Nanomaterials	Cancer	Antigen and Mechanism of Action	References
pH	HA liposomes	Lymphoma	Ovalbumin Targeting DC	[72]
	Chitosan micelles	Melanoma	Ovalbumin Targeting DC	[74]
	MGlu-HPG-modified liposomes	Lymphoma	CPG Targeting DC	[75]
	5,6-dimethylxanthenone-4-acetic acid-based micelles	Melanoma and breast cancer	Ovalbumin Activating STING pathway	[76]
	Chondroitin sulphate derived liposomes	Melanoma	Ovalbumin Targeting DC	[77]
	Caprolactone based hydrogel	Breast cancer	CPG Targeting DC	[78]
Redox	Hyperbranched poly-(amidoamine) based nanocomposite	Lymphoma	Ovalbumin Cytoplasmic delivery of antigens	[79]
pH and redox	PAMAM clusters	Pancreatic cancer	CPG Targeting draining lymph node	[80]
Hypoxia	Glycol chitosan-PEG mesoporous silica nanoparticles	Melanoma	CPG Targeting DC	[81]

**Table 4 ijms-22-12510-t004:** Exogenous stimuli-based polymer nanoparticles for delivery of antigens.

Stimuli	Polymeric Nanoparticle	Cancer Type	Antigen and Mechanism of Action	Reference
Combination PDT and hypoxia	Glycol chitosan-PEG mesoporous silica nanoparticles	Melanoma	CPG and targeting DC	[81]
Laser and ROS	Polyethyleneimine based nanoparticles	Lymphoma	OVA and targeting DC	[83]

**Table 5 ijms-22-12510-t005:** Endogenous stimuli-based polymer nanoparticles for delivery of adjuvants.

Stimuli	Polymeric Nanoparticle	Cancer Type	Mechanism of Action	References
pH sensitive	Cholesterol-DOPE-PEG based lipid nanoparticles	Lymphoma and melanoma	Activation of macrophages and plasmid DNA	[87]
ROS sensitive	poly (thioketal phosphoester) lecithin-PEG based nanoparticles	Breast cancer	Release of antigens due to LASER	[89]

**Table 6 ijms-22-12510-t006:** Endogenous stimuli-based polymer nanoparticles for delivery of agonists.

Stimuli	Polymeric Nanoparticle	Cancer Type	Agonist and Mechanism of Action	Reference
pH responsive	polymer p(DMAEMA)-b-(DMAEMA-co-BMA-co- PAA) based nanoparticle	Colon cancer	3pRNA and enhancing Anti PD L1 therapy	[96]
	mPEG-block-[DMAEMA-co-AnMA] nanocarriers	Pancreatic cancer	3pRNA and Endosomolytic carriers	[95]
	carboxyl terminated PLGA nanoshells	Colon cancer	cGMP and inducing immunogenic cell death	[97]
	PLGA nanoparticles	Lung adenocarcinoma	Small molecule 522 and antigen release due to CO2 production	[93]
Enzyme responsive	PEG vesicular nanoparticles		Imidazoquinoline and bringing Dendritic cells to Lymph nodes	[94]

**Table 7 ijms-22-12510-t007:** Exogenous stimuli-based polymer nanoparticles for delivery of agonists.

Stimuli	Polymeric Nanoparticle	Cancer Type	Agonist and Mechanism of Action	Reference
NIR II- PTT	DSPE-PEG nanoagonists	Breast cancer	Resiquimod and targeting DC	[99]
Radiation	PEG nanoparticles	Breast cancer	RGD peptide and targeting NK cells	[98]

**Table 8 ijms-22-12510-t008:** Polymers used for codelivery of antigens and adjuvants.

Antigens	Adjuvants/Agonists	Polymer	Cancer Type	Mechanism of Action	Reference
Ovalbumin	Imiquimod	PLGA	Melanoma	Gel−sol−gel transformation for DC activation	[53]
TRP-2 peptide	CPG	PCL-PEG, PCL-PEI	Melanoma	Activating DC	[90]
Ovalbumin	CL264	PEOz-PLA	Lymphoma	Activating DC	[100]

## Data Availability

Not applicable.

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
