# Peer review of "Stimuli-Responsive Polymeric Nanomaterials for the Delivery of Immunotherapy Moieties: Antigens, Adjuvants and Agonists"

_ijms, 2021, doi:10.3390/ijms222212510_

Round 1
Reviewer 1 Report
The authors present a review article on stimuli-responsive polymeric nanomaterials for the delivery of antigens, adjuvants, and agonists as cancer immunotherapies. The article is mainly divided into subsections for individual moieties and the authors describe endogenous and exogenous stimuli required for nanomaterials to respond.
Comments/suggestions
-Figures are difficult to read, especially those adopted from other sources.
-The authors use lots of acronyms but they are not well defined throughout the entire article. This made it difficult to read and follow.
-Perhaps the authors could provide in-depth explanations of immunotherapy and how antigens, adjuvants, and agonists play roles and their mechanisms.
-The authors' statement on nanotechnology in lines 38-40 needs a bit more clarity on why nanotechnology or nanomaterials are important and relevant for this particular article. The authors make similar statements throughout but did not provide adequate explanations.
-While the article includes relevant information and citations, the article can be improved if the authors could provide in-depth details of examples throughout the entire article e.g., how the particles were administered, whether it was in vitro or in vivo experiment, what animal models were used.
-Lines 195-198, the statement regarding hydrogel didn't seem to fit well. A better transition or details is needed.
-Also, the authors could provide their views and perspectives rather than just summarizing and listing advantages/disadvantages in the conclusion sections.
-Introducing the examples of each moiety before moving to stimuli subsections will be helpful. For example, TLR7/8, cGAMP agonists are described in section 3.3.1. so list these agonists in the paragraph under 3.3.
Overall organization can be improved as well, and some paragraphs and sentences seem disconnected.
Author Response
The authors thank the reviewers for their valuable comments. We have made changes according to the reviewers comments as following.
Point 1: Figures are difficult to read, especially those adopted from other sources.
Response 1: Figure 3,4 and 5 has been replaced with a better quality images.
Point 2: The authors use lots of acronyms but they are not well defined throughout the entire article. This made it difficult to read and follow.
Response 2: A separate table for Abbreviations has been made and added to the end of the review.
Point 3: Perhaps the authors could provide in-depth explanations of immunotherapy and how antigens, adjuvants, and agonists play roles and their mechanisms.
Response 3: Changes has been made throughout the article.
Point 4: The authors' statement on nanotechnology in lines 38-40 needs a bit more clarity on why nanotechnology or nanomaterials are important and relevant for this particular article. The authors make similar statements throughout but did not provide adequate explanations.
Response 4: Changes have been made throughout the article
Point 5: While the article includes relevant information and citations, the article can be improved if the authors could provide in-depth details of examples throughout the entire article e.g., how the particles were administered, whether it was in vitro or in vivo experiment, what animal models were used.
Response 5: Changes has been made to the important parts throughout the article.
Point 6: Lines 195-198, the statement regarding hydrogel didn't seem to fit well. A better transition or details is needed.
Response 6: Changes have been made throughout the article
Point 7: Also, the authors could provide their views and perspectives rather than just summarizing and listing advantages/disadvantages in the conclusion sections.
Response 7: Changes have been made throughout the accordingly.
Point 8: Introducing the examples of each moiety before moving to stimuli subsections will be helpful. For example, TLR7/8, cGAMP agonists are described in section 3.3.1. so list these agonists in the paragraph under 3.3.
Response 8: Changes has been made throughout the article.
Point 9: Overall organization can be improved as well, and some paragraphs and sentences seem disconnected.
Response 9: Changes have been made throughout the article

Reviewer 2 Report
The review shows excessive overlap with the following review:
Front. Mol. Biosci., 18 December 2020 https://doi.org/10.3389/fmolb.2020.610533 "Tumor Microenvironment-Stimuli Responsive Nanoparticles for Anticancer Therapy"
The review should be properly revised to remove overlaps and just keeping the original parts.
Author Response
Point 1: The review shows excessive overlap with the following review:
Front. Mol. Biosci., 18 December 2020 https://doi.org/10.3389/fmolb.2020.610533 "Tumor Microenvironment-Stimuli Responsive Nanoparticles for Anticancer Therapy"
The review should be properly revised to remove overlaps and just keeping the original parts.
Response 1: Thank you for your valuable comment reviewer. The mentioned review focuses on stimuli responsive nanoparticles for the anticancer therapy and the current review focuses specifically on stimuli responsive polymers in nanotechnology for immunotherapy. And we have checked the plagiarism for the same article and overlap was seen as seen in the images attached.

Reviewer 3 Report
We recommend to the authors small revisions when editing the text, as follows:
- description of all abbreviations and their subsequent use in the text, applied for: Table 1 (insert an explanatory row with the abbreviations in the table), Table 2 (insert an explanatory row with the abbreviations in the table), page 3/line 88 (GSH), etc.;
- marking Scheme 1 and Scheme 2 as Figures, resuming the numbering of the figures throughout the manuscript;
- page 3/line 88 – "GSH has an exceptional";
- p 5/158 – "pH-sensitive";
- p 6/Table 3 – "CpG", "pH & redox";
- p 7/Table 4 – "CpG";
- p 10/ Table 6 – "pH-responsive"; 'CO2";
- p 12/Table 8 – "CpG";
- for all tables and figures – main captions (bold, italics).
Author Response
The authors thank the reviewers for their valuable comments. We have made changes according to the reviewers comments as following.
We recommend to the authors small revisions when editing the text, as follows:
Point 1: description of all abbreviations and their subsequent use in the text, applied for: Table 1 (insert an explanatory row with the abbreviations in the table), Table 2 (insert an explanatory row with the abbreviations in the table), page 3/line 88 (GSH), etc.;
Response 1: A separate table for Abbreviations has been made and added to the end of the review.
Point 2: marking Scheme 1 and Scheme 2 as Figures, resuming the numbering of the figures throughout the manuscript;
Response 2: Changes has been made accordingly.
Point 3: page 3/line 88 – "GSH has an exceptional";
p 5/158 – "pH-sensitive";
p 6/Table 3 – "CpG", "pH & redox";
p 7/Table 4 – "CpG";
p 10/ Table 6 – "pH-responsive"; 'CO2";
p 12/Table 8 – "CpG";
Response 8: A separate table for Abbreviations has been made and added to the end of the review.
Point 9: for all tables and figures – main captions (bold, italics).
Response 9: Changes has been made accordingly.

Reviewer 4 Report
Prof. Yong Yeon Jeong wrote a review paper about using stimuli-responsive polymeric nanomaterials for cancer immunotherapy. The content is mainly divided into sections based on endogenous and exogenous stimulation; and the delivery of antigens, adjuvants and agonist. The article is very interesting as immunotherapy of cancer is a hot topic in oncology research. However, I feel the article is lack in depth and can be significantly improved if more content is added about the necessity or advantages of using polymeric nanomaterials in immunotherapy.
Major and minor comments are listed below:
Major
- The quality of Figure 1 needs to be improved. Figure 1 is only barely readable even on large screen. Some front size is too small and the resolution is too low. Is it possible to acquire raw figure from the original author, or maybe use similar figure from other research article with high quality.
- I feel section 2 “Stimuli-responsive polymeric nanomaterials” and section 3 “Stimuli-responsive polymers for antigens, adjuvants and agonists” are too identical. They are mainly talking about the same thing. I suggest is it possible to focus more on the immunotherapy moieties in section 3. For example: Explain the interaction between immunotherapy moieties with polymer nanoparticle. The interaction of combining immunotherapy moieties with drugs such as anti-PDL1 and DOX. Or the advantage of using polymer material in comparison with other materials in the delivery of immunotherapy moieties, etc.
- I suggest give one detail example in every subsection (2.1, 2.2, etc.) of how the nanomaterial is activate and delivers immunotherapy moieties. UV, heat, pH and etc. can activate stimuli-responsive materials is already very well known. But how do these materials response to these stimuli, and at what temperature or pH will they response is what makes the article interesting. This will highly enrich the content of the article.
Minor
- Many abbreviations are lack of their full name. Such as DOX, DCs, HA, CPG and etc., should have their full name shown at least once.
- In page 3, line 97 and 98, “The differences in cancer cells that act as endogenous stimuli are summarized in Figure 1.” I feel figure 1 is irrelevant to this. Figure 1 is talking about the releasing mechanism of pH sensitive HA gel, not about different stimuli in cancer.
- Page 8, line 224 to 230. I feel this paragraph showed be listed in section 3.2.2. Exogenous stimuli. Although the particle is ROS sensitive, but ROS response is triggered by laser, so exogenous stimuli is maybe more relevant.
Author Response
The authors thank the reviewers for their valuable comments. We have made changes according to the reviewers comments as following.
Major
Point 1: The quality of Figure 1 needs to be improved. Figure 1 is only barely readable even on large screen. Some front size is too small and the resolution is too low. Is it possible to acquire raw figure from the original author, or maybe use similar figure from other research article with high quality.
Response 1: Image has been replaced with a better quality image .
Point 2: I feel section 2 “Stimuli-responsive polymeric nanomaterials” and section 3 “Stimuli-responsive polymers for antigens, adjuvants and agonists” are too identical. They are mainly talking about the same thing. I suggest is it possible to focus more on the immunotherapy moieties in section 3. For example: Explain the interaction between immunotherapy moieties with polymer nanoparticle. The interaction of combining immunotherapy moieties with drugs such as anti-PDL1 and DOX. Or the advantage of using polymer material in comparison with other materials in the delivery of immunotherapy moieties, etc.
Response 2: Changes has been made throughout the article.
Point 3: I suggest give one detail example in every subsection (2.1, 2.2, etc.) of how the nanomaterial is activate and delivers immunotherapy moieties. UV, heat, pH and etc. can activate stimuli-responsive materials is already very well known. But how do these materials response to these stimuli, and at what temperature or pH will they response is what makes the article interesting. This will highly enrich the content of the article.
Response 3: Mechanism for each stimuli has been added with one example in sections 2.1 ND 2.2.
Minor
Point 4: Many abbreviations are lack of their full name. Such as DOX, DCs, HA, CPG and etc., should have their full name shown at least once.
Response 4: An separate table for abbreviations has been included.
Point 5: In page 3, line 97 and 98, “The differences in cancer cells that act as endogenous stimuli are summarized in Figure 1.” I feel figure 1 is irrelevant to this. Figure 1 is talking about the releasing mechanism of pH sensitive HA gel, not about different stimuli in cancer.
Response 5: It was a mistake while editing, it was supposed to be written as Scheme 1 (now captioned as figure 1 according to other reviewer’s comments)
Point 6: Page 8, line 224 to 230. I feel this paragraph showed be listed in section 3.2.2. Exogenous stimuli. Although the particle is ROS sensitive, but ROS response is triggered by laser, so exogenous stimuli is maybe more relevant.
Response 6: Agree that the ROS is stimulated by the LASER, but the LASER stimulates the photosensitizers and because of the ROS polymers are releasing drugs. The polymers are ultimately stimulated by the ROS. Since the review is about polymers that are responsive to stimuli the section has been kept at that particular section.

Round 2
Reviewer 1 Report
The authors addressed all concerns.
Reviewer 2 Report
The work is well presented, publication is recommended in IJMS
Reviewer 4 Report
Prof. Yong Yeon Jeong and his colleges have done a good job improving the review paper to a higher level. The revise version has a lot more content on the releasing and interaction mechanism of polymeric materials, as well as on giving examples of how the drugs are released on each kind of carriers. These contents highly increase the depth of the article. Minor editing is needed on English spelling and grammar, which I believe can be done by English editing service of the journal. The quality of Figure 3 is still not very good. Only the figure is enlarged, but the resolution is still low. If possible, it will be better to redraw some parts of the figure, or use similar figure with better quality.
Still over all, good improvement on the article.